# Technical Note: Stability of tris pH buffer in artificial seawater stored in bags

Wiley H. Wolfe[1], Kenisha M. Shipley[1], Philip J. Bresnahan[2], Yuichiro Takeshita[3], Taylor Wirth[1], Todd R. Martz[1]

[1]Scripps Institution of Oceanography, University of California San Diego, La Jolla, 92093, USA
[2]Department of Earth and Ocean Sciences, University of North Carolina Wilmington, Wilmington, 28403, USA
[3]Monterey Bay Aquarium Research Institute, Moss Landing, 95093, USA

*Correspondence To:* Philip J. Bresnahan Jr. (bresnahanp@uncw.edu)

**Abstract**
Equimolal tris (2-amino-2-hydroxymethyl-propane-1,3-diol) buffer in artificial seawater is a well characterized
and commonly used standard for oceanographic pH measurements. We evaluated the stability of tris pH when stored
in purportedly gas impermeable bags across a variety of experimental conditions, including bag type, and storage in
air vs. seawater over 300 days. Bench-top spectrophotometric pH analysis revealed that the pH of tris stored in bags
decreased at a rate of $0.0058 \pm 0.0011$ $yr^{-1}$ (mean slope $\pm$ 95% confidence interval of slope). The upper and lower
bounds of expected pH change at t = 365 days, calculated using the averages and confidence intervals of slope and
intercept of measured pH change vs. time data, were -0.0042 and -0.0076 from initial pH. Analyses of total dissolved
inorganic carbon confirmed that a combination of $CO_2$ infiltration and/or microbial respiration led to the observed
decrease in pH. Eliminating the change in pH of bagged tris remains a goal, yet the rate of pH change is lower than
many processes of interest and demonstrates the potential of bagged tris for sensor calibration and validation of
autonomous in situ pH measurements.
**1.    Introduction**
Ocean pH is a key measurement used for tracking biogeochemical processes such as photosynthesis,
respiration, and calcification (Takeshita et al., 2016); and represents perhaps the most recognized variable associated
with ocean acidification (OA), the decrease in ocean pH due to the uptake of anthropogenic carbon dioxide (Doney et
al., 2009). OA progresses with a global average pH decline of 0.002 per year in the surface open ocean (Bates et al.,
2014), and the accumulated and projected near-term effects of OA have been shown to have deleterious effects on
many calcifying organisms (Cooley and Doney, 2009). Beyond the narrow scope of calcifiers, organismal response is
complex, exhibiting varied responses across processes such as reproduction, growth rate, and sensory perception.
Organismal responses are further complicated by their impact on ecosystem level dynamics, such as altering
competition and predator-prey relationships (Doney et al., 2020). Furthermore, pH effects are often exacerbated by
concomitant stressors, such as decreased dissolved oxygen or increased temperature. Ultimately, OA will affect
humans through impacts on fisheries, aquaculture, and shoreline protection (Branch et al., 2013; Doney et al., 2020).
The quality of pH measurement required to observe various phenomena is often broken into "climate" and
"weather" levels of uncertainty (Newton et al., 2015), or 0.003 and 0.02, respectively. Discrete sampling has been
shown to be capable of meeting the climate level of uncertainty when best practices are followed, yet many labs do
not consistently meet this standard (Bockmon and Dickson, 2015). Furthermore, while discrete, bench-top
methodologies can be the most accurate, the ocean's vast size limits the oceanographic community's ability to make
ship-based discrete pH measurements to decadal reoccupations of a few major sections per ocean basin (Sloyan et al.,
2019). The sparsity of ship-board measurements hinders our ability to assess sub-decadal processes, such as seasonal
cycles or bloom events, over much of the ocean (Karl, 2010), and highlights the need for autonomous, high-frequency
pH measurements. Technological advancements have led to more routine autonomous pH measurements over the past
decade, providing opportunities to fill some gaps in time and space in discrete sampling programs (e.g, Byrne, 2014;
Martz et al., 2015; Lai et al., 2018; Wang et al., 2019; Tilbrook et al., 2019). Globally, pH sensors now operate on
hundreds of autonomous platforms including moorings and profiling floats, delivering unique datasets in the form of
Eulerian and depth resolved Lagrangian time series (Johnson et al., 2017; Bushinsky et al., 2019; Sutton et al., 2019).
While sensors increase data coverage, many sensor-based pH measurements, particularly on moored systems, continue
to fall short of both climate and weather levels of uncertainty, as highlighted in the intercomparison tests carried out
by the Alliance for Coastal Technologies (ACT, 2012) and by the Wendy Schmidt Ocean Health XPRIZE (Okazaki
et al., 2017).
Independent validation is typically required for autonomous sensors to meet both weather and climate levels
of uncertainty. For example, autonomous underway $p$CO$_2$ systems (Pierrot et al., 2009), moorings (Bushinsky et al.,
2019), and autonomous surface vehicles (Chavez et al., 2017; Sabine et al., 2020) are able to provide climate quality
observations with an uncertainty of $\pm 2$ µatm because traceable standard gases are frequently measured in situ. For pH
measurements on profiling floats (Johnson et al., 2016), sensor performance is validated by comparing to a deep
reference pH field that is calculated using empirical algorithms (Williams et al., 2016; Bittig et al., 2018; Carter et al.,
2018). This approach has demonstrated the ability to obtain high quality pH measurements from a network of profiling
floats (Johnson et al., 2017) but requires measurements in the deep ocean where pH is comparatively stable. It is
atypical for other pH sensors, including coastal moored sensors, to have an automated or remote validation. Therefore,
on such deployments, validation has largely relied on discrete samples taken alongside the sensor (Bresnahan et al.,
2014; McLaughlin et al., 2017; Takeshita et al., 2018), which presents unique challenges; primarily that spatiotemporal
discrepancy can lead to errors of $> 0.1$, especially in highly dynamic systems (Bresnahan et al., 2014).
Similar to the method in use by $p$CO$_2$ systems, one approach to validate in situ pH sensors is by measuring a
reference material or pH standard, repeatedly during a sensor deployment. The most commonly used standard for
oceanographic pH measurement is equimolal tris (2-amino-2-hydroxymethyl-propane-1,3-diol) buffer in artificial
seawater (ASW), hereafter referred to as tris or tris-ASW (DelValls and Dickson, 1998; Papadimitriou et al., 2016).
The pH of tris has been characterized over a range of temperature, salinity, and pressure (DelValls and Dickson, 1998;
Rodriguez et al., 2015; Takeshita et al., 2017; Müller et al., 2018), allowing for accurate calculation of tris pH across
a wide range of marine conditions. Furthermore, when stored in borosilicate bottles and under ideal conditions, these
buffers have been shown to be stable to better than 0.0005 over a year (Dickson, 1993; Nemzer and Dickson, 2005),
making tris a good candidate for in situ validation of long-term deployments of autonomous pH sensors. To be utilized
for in situ applications, the reference solution must be stored in bags (as in, Hales et al., 2005; Seidel et al., 2008;
Sayles and Eck, 2009; Spaulding et al., 2014; Wang et al., 2015; Lai et al., 2018). Recently, in situ sensor validation
using bagged tris was demonstrated by Lai et al. (2018) during a 150-day deployment of an autonomous pH sensor,
where the tris standard was measured in situ every 5 days. However, the stability of tris when stored in bags has not
been quantified systematically using spectrophotometric bench-top pH measurement techniques recommended as best
practices (Dickson et al., 2007).
In this work we quantified the stability of tris stored in bags for 300 days. Tris from four separately prepared
batches was stored in two bag types either in a lab or submerged in seawater. In addition, one batch was stored in
borosilicate bottles in the lab as a control. Spectrophotometric pH measurements were made approximately every two
months on each bag of tris. Throughout the experiment, Certified Reference Materials (CRMs) for oceanic $CO_2$
measurements (Dickson, 2001) were used to assess the stability of the spectrophotometric pH system.

## 2.    Methods

Two bag types were tested for storing tris (Figure 1). Bag type 1 was custom made based on a design used in
the "Burke-o-Lator" system (Hales et al., 2005; Bandstra et al., 2006), made from PAKDRY 7500 barrier film
(IMPAK P75C0919). The barrier film is made of layers of polyester and nylon with a sealant layer of metallocene
polyethylene. Two 23 x 48 cm (9" x 19") sheets were heat sealed on three sides, forming a pocket, and a 1.9 cm (¾")
diameter hole was cut into one of the pocket walls for the bulkhead fitting and bulkhead nut (McMaster-Carr
8674T55). The bulkhead was sealed into the wall with a silicone gasket (McMaster-Carr 9010K13), washer
(McMaster-Carr 95649A256), and coated with silicone sealant (McMaster-Carr 74955A53). A "push-to-connect" ball
valve fitting (McMaster-Carr 4379K41), was attached to the bulkhead. The modified pocket was rinsed, dried, and
heat sealed along the final edge to create a ~4 L bag. Bags were left to dry for at least 24 hours before filling. Bag type
2 was a commercially available 3 L Cali-5-Bond bag purchased from Calibrated Instruments and used without
modification. It is a multi-layer bag made of plastic, aluminium foil (to prevent liquid and gas permeation), a layer of
inert high density polyethylene (to form a non-reactive inner wall) and, a polycarbonate Stopcock Luer valve.

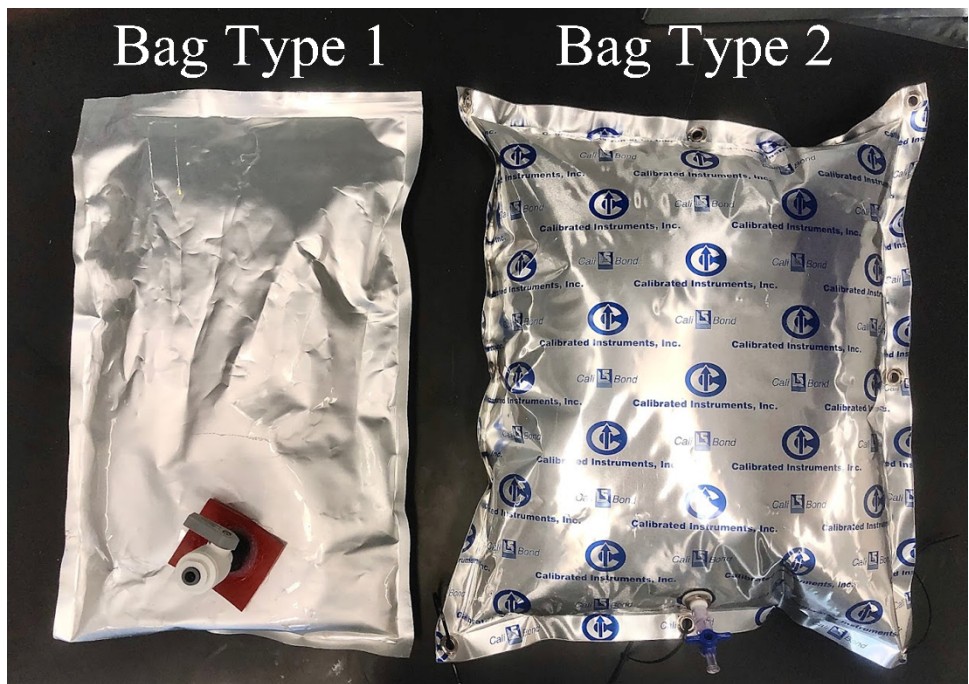

**Figure 1: A picture of bag type 1 and 2 used to store tris in this study.**

In this experiment, four batches of tris were prepared following the procedure in DelValls and Dickson
(1998), using off-the-shelf reagents with no additional standardization or purification (e.g. recrystallization of salts).
The focus of this paper is stability of bagged tris over time and does not prioritize obtaining highly accurate equimolal
tris (as would be necessary for characterization of thermodynamic constants, for example). The calculated pH of tris

in this study was 8.2652 at 20°C, based on quantity of reagents used. This is 0.0135 higher than the pH of equimolal

tris, 8.2517 at 20°C (DelValls and Dickson, 1998). The pH discrepancy was due to a unit error in the measurement of

HCl (our preparation used mol/L rather than the prescribed mol/kg-sol). This unit error resulted in a tris:trisH$^+$ of

1:0.97 that slightly differs from the 1:1 of truly equimolal tris. As this ratio is nearly equimolal, the term "equimolal"

will continue to be used throughout this study. The details of the specific reagents used to prepare the tris solution can

be found in Table A1.

Three stability tests were initiated at different times over the course of 18 months. The initiation of a given

test is defined as the date of preparation of the tris used in that test. A summary of the differences between these tests

is shown in Table 1 and described here. Each bag has a unique identifier in the format of "Batch #, Bag #, Lab or

Tank." If this identifier is duplicative, the bags are differentiated with letters A to D. Each bag was rinsed before

filling: 3 times with deionized water (DI), 5 times with ultrapure water (> 18 MΩ resistivity) and at least 3 times with

200 mL of tris. Tris bags were stored on a lab bench or in a 5,000 L test tank filled with ozone-sterilized, filtered

seawater. Bag type 2 experienced delamination of exterior layers when stored in seawater during test 2 and was not

used in further testing. Tris from batch 4 was also stored in borosilicate bottles following the procedure in Nemzer

and Dickson (2005). In addition to pH measurements, dissolved inorganic carbon ($C_T$) was measured on both bagged

and bottled tris during test 3 to see if changes in pH were due to increased $CO_2$. $C_T$ samples were measured using a

custom-built system based on an infrared (IR) analyser (LI-COR 7000) similar to systems used by O'Sullivan and

Millero (1998) and Friederich et al. (2002). This IR measurement system is capable of measuring relatively low $C_T$

without requiring method adjustment and has been used to make near zero $C_T$ measurements (Paulsen and Dickson,

unpublished data). $C_T$ measurements were made on CRMs (Batch 179 & 183). The precision of the $C_T$ measurements

was ± 1.4 μmol/kg (pooled standard deviation, $n_{samples}$=15, $n_{measurements}$=44).

**Table 1: Tris preparation and storage.**

| | Bag Type | Tris Batch | Date Made | Storage Location | Rinse Procedure | $C_T$ Measured |
|---|---|---|---|---|---|---|
| Test 1 | 1 & 2 | 1 & 2 | 13 Dec 2017 | Lab & Tank | 3x DI, 5x ultrapure, 3x tris | No |
| Test 2 | 1 & 2 | 3 | 13 April 2018 | Lab & Tank | 3x DI, 5x ultrapure, 3x tris | No |
| Test 3 | 1 & bottle | 4 | 26 February 2019 | Lab | 3x DI, 5x ultrapure, ≥ 6x tris | Yes |

Tris pH was measured every 55 ± 20 days (mean ± standard deviation of measurement interval) throughout

the experiment. The pH of tris was measured in triplicate at each time point with spectrophotometry using m-cresol

purple as the indicator dye using the system described in Carter et al. (2013). Absorbance measurements were made

in a 10-cm jacketed cell, and the temperature was measured directly adjacent to the cell outflow using a NIST-traceable

thermometer (± 0.1 °C, QTI DTU6028P-001-SC). Blank and sample were held for 3 minutes in the jacketed flow cell

prior to absorbance measurements.

On average, temperature was stable to within a 0.02 °C range over the course of the day; the mean temperature
throughout the experiment was 20.09 ± 0.23 °C (1 σ), although temperature was 0.6 °C higher than the average on
one measurement day. Spectrophotometric pH measurements are reported at 20 °C by adjusting the measured pH
value at the measured cell temperature $T_C$ ($pH_{spec,T_C}$) to 20 °C ($pH_{spec,20°C}$) using the known temperature dependence
of tris ($pH_{tris}$) as follows:

$$pH_{spec,20°C} = pH_{spec,T_C} - (pH_{tris,T_C} - pH_{tris,20°C}) \qquad (1)$$

$pH_{tris,T_C}$ and $pH_{tris,20°C}$ are the theoretical pH of tris (at the measured temperature and 20 °C respectively) and were
calculated using Eq. (18) in DelValls and Dickson (1998). This adjustment assumes that any potential difference in
$\partial pH/\partial T$ between that corresponding to equimolal tris and that corresponding to our 1:0.97 tris:trisH$^+$ ratio has a
negligible effect over the small temperature range observed.
To account for pH-dependent errors from impurities in unpurified mCP, a pH-dependent correction factor
was determined based on the protocol outlined in Takeshita et al. (*in review*). Briefly, pH of natural seawater with
different ratios of added tris:trisH$^+$ was measured subsequently using impure dye ($pH_{impure}$; from Aldrich, lot
MKBH6858V) and purified dye ($pH_{pure}$; from Robert Byrne's laboratory, University of South Florida (Liu et al.,
2011)) over a range of pH between 7.4 to 8.2 at approximately 0.2 intervals. Varying ratios of tris:trisH$^+$ were used to
obtain different solution pH, and to buffer any changes in pH during the experiment, which negates the need for dye
perturbation corrections in this characterization. Triplicate measurements were made at each pH. A second order pH-
dependent error was observed as previously described, following the equation ($R^2 = 0.975$, RMSE = 0.000434):

$$pH_{pure} = -0.0047777 \times pH_{impure}^2 + 1.0668875 \times pH_{impure} - 0.2359740 \qquad (2)$$

All subsequent $pH_{spec}$ measurements in this study were conducted with impure dye and are reported with this dye
impurity correction (Eq. 2) applied. The correction adjusted the reported pH by 0.0093 ± 0.0002 (mean ± standard
deviation, n = 126). No dye perturbation correction was used (a correction for a change in pH caused by the addition
of the dye). As the high buffering capacity of tris, in combination with a dye adjusted to a pH similar to that of tris,
results in a negligible change in measured pH.
Measurements of tris batches 1 and 2 made in the first 150 days have been removed from the data set due to
procedural changes made to the spectrophotometric pH system to correct for problems with temperature equilibration.
Outliers were removed from the spectrophotometric pH measurements if the absorbance at 760 nm was above 0.005
or below –0.002 (indicative of a measurement problem, such as a bubble or lamp drift), resulting in the removal of 2
out of 163 measurements. Additionally, outliers were removed from the data set if they were greater than three
standard deviations from the mean of a measurement triplicate, where standard deviation is calculated as using all sets
of triplicates (1 standard deviation = 0.0004, n = 55), resulting in the removal of 2 of 161 remaining measurements.
The remaining 159 measurements were used for the analysis presented here. An analysis of variation, or ANOVA,
was used to detect the dependence of the results on tris batch, bag/bottle type and storage location. Analysis was
performed using MATLAB R2020a and the standard function "anovan()." Throughout the experiment, CRMs
(procured from A. Dickson, Scripps Institution of Oceanography) for seawater $C_T$ and total alkalinity were measured
regularly to verify instrument performance (Dickson, 2001). A time-series of CRM measurements over the duration
of the work described here showed no systematic drift. (Fig. A1 in Appendix A).To assess if the change in pH was
driven by the addition of $CO_2$, the final pH and available $C_T$ measurements were compared with a model described
here. The theoretical change in tris-artificial seawater (ASW) pH due to an increase in $C_T$ is straightforward to
calculate, since both tris and $CO_2$ acid-base equilibria are well-characterized in seawater and ASW media. The pH is
calculated for tris-ASW + $C_T$ using an equilibrium model following the approach described in Chapter 2 of Dickson
et al. (2007) for the case of known alkalinity and $C_T$. In the case of ASW, the seawater equilibrium constants for $CO_2$
are appropriate because minor ions present in seawater and not ASW do not appreciably affect the $CO_2$ equilibrium
constants (particularly when the goal is to compute relative changes in pH) as the ionic background of ASW is closely
matched to that of seawater at salinity = 35. In our model, minor acid-base species important to seawater alkalinity
but not present in ASW (borate, phosphate, silicate, fluoride) are set to zero. The definition of total alkalinity is
modified to include the tris acid-base system following the definition of acid-base donor/acceptor criteria given by
Dickson (1981): tris is assigned as a level-1 proton acceptor and tris-$H^+$ is at the zero level. Thus, in our model, $tris_{tot}$
= 0.08 molal and alkalinity = 0.04 molal and $C_T$ is a variable. An algorithm (see Annexe 1 in Dickson et al. (2007)) is
then used to find the root of the alkalinity equation in its residual form by solving for pH.
**3.    Results & Discussion**
Figure 2 depicts $pH_{spec,20°C}$, stored in either a bag or bottle, as a function of time and is subdivided for tests 1,
2, and 3. A linear decrease was observed for all bags or bottles. A linear regression was calculated for each
experimental condition and, in the cases where measurements at t = 0 were removed due to protocol changes described
above, the line was extrapolated back to t = 0, shown by the dotted line. The measured or extrapolated y-intercept is
reported as the initial pH in Table 2. In all tests, trendlines are extrapolated to t = 365 days to illustrate observed and
predicted change over the course of a year as shown by the solid line. For ease of visual comparison, the y-axis of
each subplot has an identical pH range of 0.017.

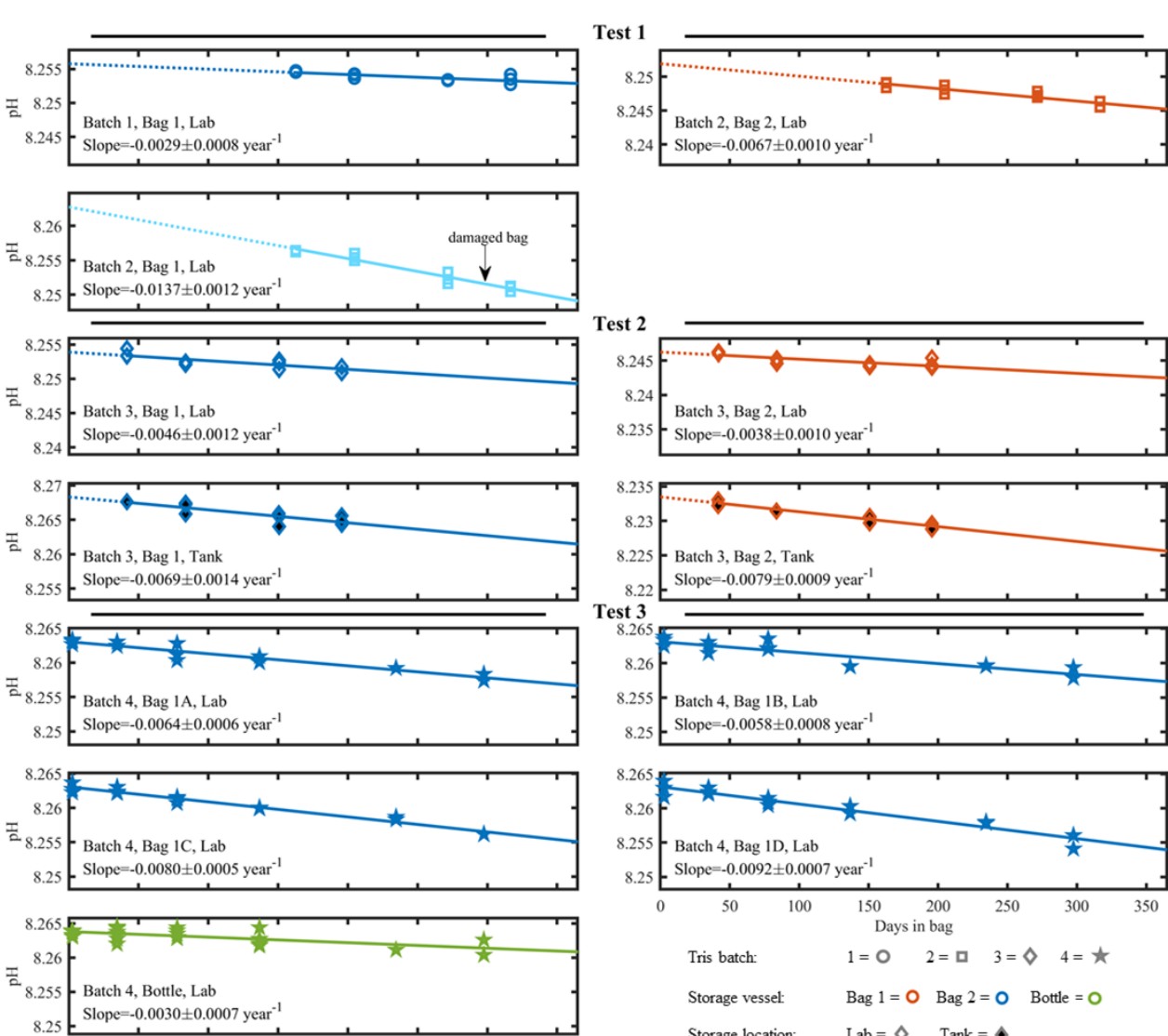

**Figure 2: Individual time series of measured pH in tris buffer solutions. Tris batch is indicated by shape, storage vessel by**
**color, and storage location by fill. This marker system is also followed in Fig. A2. The solid line is a linear regression starting**
**at the first included pH measurement and ending 365 days after the tris was bagged. The dotted line illustrates the**
**extrapolation back to 0 days stored in bag when measurements at t = 0 do not exist. The range of the y-axis scale is fixed at**
**0.017 pH for all subplots.**

**Table 2: Linear regression statistics from trendlines shown in Fig. 1 and 2. The last row shows the regression statistics for tris from all batches, in either bag type, stored in the lab or test tank. Slope and intercept are shown as mean ± 95% confidence intervals. The reported intercept is the regression intercept; when initial pH measurements are available, they differ by less than 0.0003 from regression intercept. \* Indicates the outlier (Batch 2, Bag 1, Lab) caused by a damaged bag. The outlier, "Batch 2, Bag 1, lab", was not used in the "All Batches, All Bags, Lab or Tank" composite. † In all batches, all bags, lab or tank, the slope was calculated with a linear fit of all (non-outlier) tris measurements. The RMSE is the mean RMSE of all (non-outlier) bag fits. ‡ The calculated tris pH was calculated at 20°C; however, this calculated pH is 0.0135 higher than equimolal tris as noted above (DelValls and Dickson, 1998).**

| Batch & Storage Method | Slope (mpH yr$^{-1}$) | Intercept (Initial pH) | RMSE (mpH) | $r^2$ | n |
|---|---|---|---|---|---|
| Batch 1, Bag 1, Lab | -2.9 ± 1.7 | 8.2558 ± 0.0012 | 0.43 | 0.59 | 12 |
| Batch 2, Bag 1, Lab* | -13.7 ± 2.7 | 8.2627 ± 0.0018 | 0.61 | 0.94 | 11 |
| Batch 2, Bag 2, Lab | -6.7 ± 2.2 | 8.2519 ± 0.0015 | 0.55 | 0.82 | 12 |
| Batch 3, Bag 1, Lab | -4.6 ± 2.7 | 8.2539 ± 0.0010 | 0.62 | 0.62 | 11 |
| Batch 3, Bag 1, Tank | -6.9 ± 3.2 | 8.2683 ± 0.0012 | 0.73 | 0.73 | 11 |
| Batch 3, Bag 2, Lab | -3.8 ± 2.1 | 8.2462 ± 0.0008 | 0.54 | 0.61 | 12 |
| Batch 3, Bag 2, Tank | -7.9 ± 2.1 | 8.2335 ± 0.0008 | 0.44 | 0.92 | 9 |
| Batch 4, Bag 1A, Lab | -6.4 ± 1.3 | 8.2630 ± 0.0005 | 0.64 | 0.90 | 14 |
| Batch 4, Bag 1B, Lab | -5.8 ± 1.8 | 8.2631 ± 0.0008 | 0.91 | 0.79 | 15 |
| Batch 4, Bag 1C, Lab | -8.0 ± 1.0 | 8.2631 ± 0.0004 | 0.49 | 0.96 | 15 |
| Batch 4, Bag 1D, Lab | -9.2 ± 1.6 | 8.2631 ± 0.0007 | 0.80 | 0.92 | 15 |
| Batch 4, Bottle, Lab | -3.0 ± 1.4 | 8.2638 ± 0.0005 | 0.81 | 0.44 | 25 |
| All Batches, All Bags, Lab or Tank† | -5.8 ± 1.1 | – | 0.72 | 0.66 | 126 |
| Calculated tris pH‡ | – | 8.2652 | – | – | – |

Only bags from test 3, using tris batch 4 and bag type 1, have direct initial pH measurements and replicate bags. Initial pH measurements of these 4 bags were 8.2630 ± 0.0007 (mean ± standard deviation, n = 12). Importantly, the very low standard deviation suggests that a single initial pH measurement is representative of all replicate bags filled with a single tris batch, if the preparation procedure used in test 3 is followed. This inter-bag consistency is beneficial because it reduces the number of initial pH measurements required when filling multiple bags. There is also strong agreement in initial pH measurements between bagged and bottled tris in test 3, with the initial pH of bottled tris 0.0007 higher than bagged tris (8.26327 ± 0.0004, n = 6). The differences in filling procedure or impurities between bags and bottles in test 3 appear to have little effect on the initial pH. The mean initial pH of tris batch 4 is 0.002 (n = 5) lower than calculated pH$_{tris,20°C}$ (Fig. A2). This difference between the mean initial pH of tris batch 4 and calculated pH$_{tris,20C}$ is similar in direction and magnitude to those reported in other studies: DeGrandpre et al. (2014) reported –0.0012 ±0.0025 and Müller and Rehder (2018) reported -0.002 to -0.008 (measured pH minus pH$_{tris,T_C}$). With standard laboratory equipment and off-the-shelf reagents, an uncertainty of 0.006 is expected in prepared tris (Paulsen and Dickson, 2020). Measurements were also made on Dickson standard tris (batch T35) using the same instrument and the pH was 0.0019 higher than the calculated pH$_{tris,20°C}$ (n = 2). In tests 1 and 2, the initial pH was extrapolated from a linear regression. The extrapolated initial pH values are more variable and lower (on average) than those directly

measured (Fig. A2). These differences may be a result of the extrapolation or different experimental variables such as
the increased rinsing of bags, or the single bag type and storage location used in test 3.

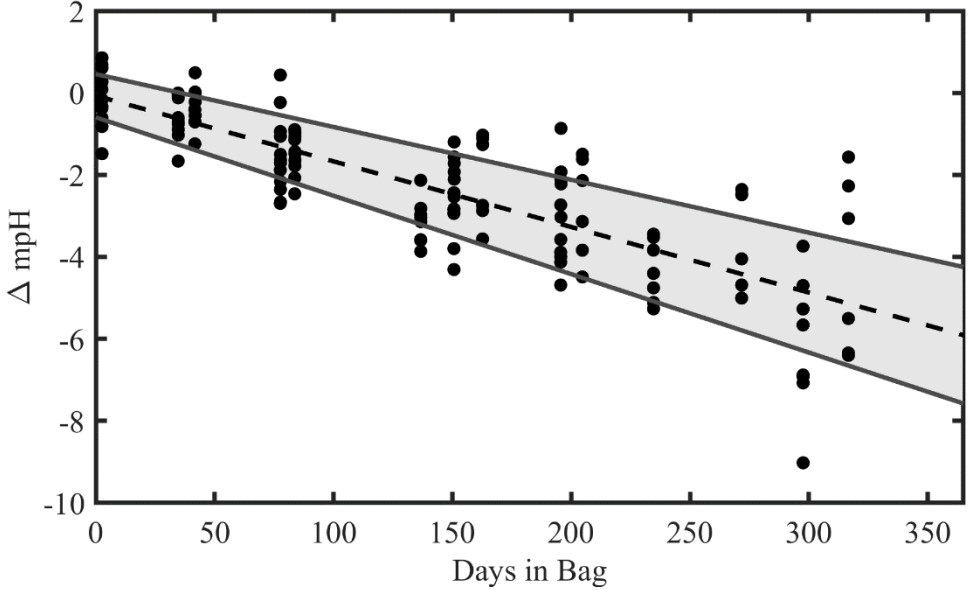


**Figure 3: Combined time series of measured pH in tris buffer tris buffer solutions. The dots represent every**
**measurement made on a (non-damaged) bag of tris. The dotted line is the "All Bags, All Batches, Lab or Tank"**
**regression. The grey shaded region is the observational 95% confidence interval (CI). The CI is intended to**
**estimate the future pH of a tris bag (with known initial pH and an unmeasured bag specific rate of change).**
**The upper and lower bounds are -0.0042 and -0.0076 pH per year, respectively.**

Figure 3 depicts a composite of all test results as the change from the initial pH of tris
$\left(\Delta pH = pH_{spec,20°C}^{t=day} - pH_{spec,20°C}^{t=0}\right)$ as a function of time elapsed since bagging. A linear regression on all pH
measurements, excluding the outlier of "Batch 2, Bag 1, Lab", of tris stored in bag types 1 or 2, has a slope of –0.0058
$\pm$ 0.0011 yr$^{-1}$ (mean $\pm$ 95% C.I.). The upper and lower bounds of $\Delta pH$ at t = 365 days, -0.0042 and -0.0076, are
important to consider when utilizing this bagged storage method of tris. These bounds provide the broadest expected
range in pH change over a year of storage and include both the intercept and slope confidence intervals (*slope_{CI}* and
*intercept_{CI}*, respectively). For example, the upper bound of $\Delta pH$ at t = 365 days is calculated as: *upper bound =*
$(slope + slope_{CI}) * 365 + intercept + intercept_{CI}$. The outlier (Batch 2, Bag 1, Lab) was excluded due to
noticeable damage to the bag (see Fig. A3 in Appendix A), which is believed to have caused its pH to decrease at
more than two times the average rate of the other bags. The damage appears to be a break in the metallic bag layer,
potentially caused by creasing or pinching of the bag when handling. This observation highlights the importance of
maintaining bag integrity, particularly during use in the field. A successful two-week field deployment has been
conducted using the tris bags described here and a modified SeapHOx in a shallow, coral reef flat (Bresnahan et al.
2021). This two-week deployment was significantly shorter than the year of storage described here and further field
testing in longer deployments in varied environments are required before widespread use of this technology. For the
longer time frame depicted in Figure 3, the only comparable example found in the literature is the work of Lai et al.
(2018). In this work, Lai et al. (2018) used bagged tris for sensor calibration, with in situ tris measurements made over
150 days. Lai et al. (2018) did not report a change in the pH of bagged tris over the deployment; however, the reported
precision of the SAMI-pH in situ instrument (± 0.003) would not resolve the expected change shown in our Figure 3.
Therefore, the results of Lai et al. (2018) are not inconsistent with our study.

A significant increase in $C_T$ was observed for all types of bags and bottles in Experiment 3 (Figure 4). A high

correlation between solution pH and $C_T$ was observed, with a slope of $-0.0029 \pm 0.0006$ pH per 100 µmol kg$^{-1}$ (n =
14, r$^2$ = 0.70), suggesting that the change in tris pH and $C_T$ was primarily driven by an increase in $CO_2$. The observed
slope agrees closely with a theoretical model prediction of a linear decrease in pH of $-0.0024$ per 100 µmol kg$^{-1}$ of
$C_T$ added (over the range of $C_T$ observed). There are two possible sources of the increasing $C_T$: gas exchange of $CO_2$
with the environment and microbial respiration within the storage vessel. Gas exchange should not be a significant
source of $CO_2$ for tris stored in a borosilicate bottle, as this is the standard equipment used to store seawater $CO_2$ and
tris buffers and is known to minimize gas exchange (Dickson et al. 2007). Therefore, it is likely that respiration was
the primary driver for the increase in $C_T$ for tris stored in bottles. On average, pH decrease of tris stored in bags was
larger than that in the standard bottle (Figure 2), indicating either an additional source of $CO_2$ from gas exchange, or
larger amounts of respiration. Distinguishing between these two theorized sources would require measurements of
additional parameters such as dissolved organic carbon.

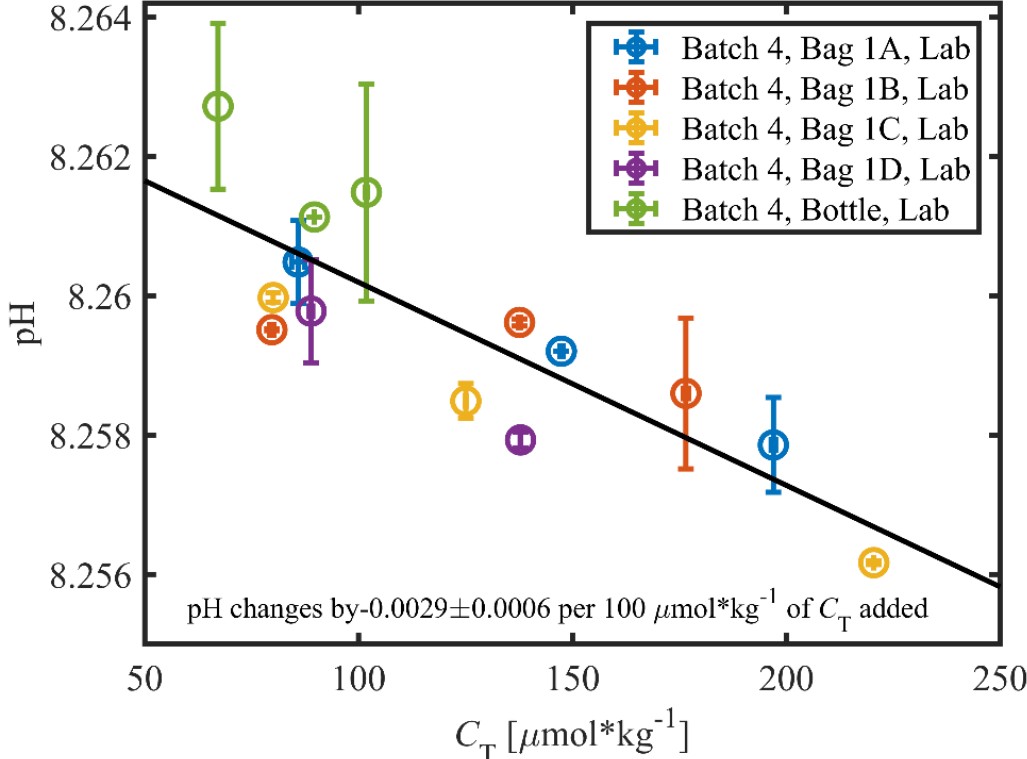


**Figure 4: pH plotted against $C_T$ shows a linear relationship between the two parameters in tris buffer with a slope of –**
**0.0029 pH for every 100 µmol kg$^{-1}$ of $C_T$ added. The measurements shown are from three sampling occurrences between**
**130–300 days stored on bags and bottles used in Test 3. Only two measurements are shown for "Batch 4, Bag 1D, Lab"**
**because it ran empty before $C_T$ were made.**
The pH stability of tris could be improved by reducing either likely source of $C_T$: gas exchange or microbial
respiration. For bags, $CO_2$ may diffuse through the fittings, gasket, or bag walls, particularly if damaged. The relatively
small breaks in the aluminium foil layer caused "Batch 2, Bag 1, Lab" to decrease more than twice as fast as the
average bag. Storage bag, fitting, and gasket material, as well as careful handling, are therefore important factors in
minimizing gas exchange. For example, silicone is permeable to $CO_2$, and thus could have been a path of gas exchange
into the tris for this experiment. As noted above, Nemzer and Dickson (2005) found an almost negligible change of
0.5 mpH $yr^{-1}$ in bottled tris. Our bottled tris changed at –3.0 mpH $yr^{-1}$ (n = 10 bottles measured over 161 days),
approximately half the rate of the tris stored in bags. While –3.0 mpH $yr^{-1}$ is near the detection limit of our
measurements, it suggests that the bottling protocol used in this study was not as well controlled as that of Nemzer
and Dickson (2005). For example, the Dickson lab at Scripps Institution of Oceanography regularly uses an annealing
oven to combust all trace organic films that may persist on glass bottles, but in our study, bottles were not annealed.
Although bags cannot be annealed, future steps that may be worth consideration to reduce microbial respiration in
bags include addition of a biocide to the tris solution, acid cleaning the bags, and using ultraviolet light to remove
organics from the ultrapure water used to prepare tris. There are some disadvantages to these proposed steps. Addition
of a biocide may not be ideal for use in sensitive environments if the tris is discharged after use and would alter the
composition of the solution slightly. While rinsing or prolonged soaking of the bags with an acid may help to remove
organics, it is unclear if it would have negative effects on the integrity of the bags. Beyond removing organics on the
bag surfaces, care should be taken to avoid introducing organic contaminates into the tris during the solution
preparation and bag filling procedures to minimize future respiration.
Both bag type 1 and 2 experienced problems with structural integrity during this experiment. A single type 2
bag experienced delamination of exterior bag layers when stored submerged in seawater, causing the eventual tearing
and failure of the bag when handling. Bag type 2 was not used in test 3 due to this failure. It should be noted that in
other studies which successfully used bag type 2, the bag was submerged in seawater for less time than in this
experiment (Sayles and Eck, 2009; Aßmann et al., 2011; Wang et al., 2015). A single bag type 1 had the subtler
problem of small breaks in the aluminium foil bag layer, likely causing an increased pH rate of change. In non-
damaged bags, factors such as bag type/bottle, lab/tank storage, or tris batch did not have statistically significant ($p$-
value < 0.05) correlations with the pH change of tris ($p$-values 0.12, 0.11 and 0.09, respectively). The results of the
ANOVA support that tris can be held in bag type 1 or 2 and stored in a lab or tank and the pH will change similarly
regardless of storage method for up to 300 days. Additional bag types could be tested, such as bags made by Pollution
Measurement Corp. used by Lai et al. (2018) or Scholle DuraShield used by Takeshita et al. (2015).
These results suggest that when bags are carefully handled prior to and after filling, tris pH changes are small
over time. Specific recommendations for further work include: bags must be handled with care and enclosed in
protective containers to prevent damage, bags must be rinsed with tris prior to filling, and additional testing is merited
to determine sources of and methods to reduce contamination, such as acid washing.

## 4. Conclusions

This article describes our characterization of the stability of tris buffer in artificial seawater when stored in purportedly gas-impermeable bags. Several different tests, initiated over the course of a year and a half and lasting up to 300 days, exhibited an average decrease of 5.8 mpH yr$^{-1}$. In comparison, tris stored in standard borosilicate bottles was shown to have a decrease of 3.0 mpH yr$^{-1}$. For yearlong deployments, an expected pH change of $-0.0058$ is well below the weather quality threshold of 0.02 pH units. This low rate of change demonstrates the value of bagged tris for in situ validation of autonomous pH sensors (regardless of sensor operating principles), particularly in highly dynamic areas where repeatability of calibration based on discrete samples is challenging. Given the thorough characterization of tris over wide ranges of environmental variables, this contribution can aid in the traceability and intercomparability of pH sensor measurements. While valuable at the current stage of development (as demonstrated by, e.g., Lai et al. (2018) and Bresnahan et al. (2021)), further development would ideally result in a commercially available bag and filling procedure that can yield a rate of pH change less than the climate threshold of 0.003 per year. This will require further tests to identify the source of $CO_2$, gas exchange or microbial respiration, as well as steps to reduce or eliminate these sources.

Periodic measurement of bagged tris in situ would allow for detection of sensor drift. Most in situ pH sensors are deployed in the euphotic zone in coastal areas, typically resulting in expedited biofouling and sedimentation, and leading to sensor drift (Bresnahan et al., 2014) that could be identified and potentially corrected. Such periodic calibration/validation would aid in identifying sensor issues and allow for greater consistency and continuity between a timeseries and planned or vicarious crossovers where an automated calibration can be used to augment or replace pre- and post-deployment calibrations/validations.

**Appendix A**

**Table A1. Detailed information about the specific reagents used to make the tris solution. *Reagent chemicals that meet or**
**surpass specifications of the British Pharmacopeia (BP), European Pharmacopeia (EP), Food Chemicals Codex (FCC),**
**United States Pharmacopeia (USP).**

| Chemical | Manufacture | Part Number | Lot Number | Batch | Assay | Grade |
|---|---|---|---|---|---|---|
| tris | Fisher Scientific | T395-1 | 170360 | all | 99.8% | Certified ACS |
| NaCl | Fisher Scientific | S641-212 | 127252 | all | 99.0 to 100.5% | *BP/EP/FCC/USP |
| $Na_2SO_4$ | Fisher Scientific | S421-1 | 134837 | all | 99.8% | Certified ACS |
| KCl | Fisher Scientific | P217-500 | 174416 | all | 99.7% | Certified ACS |
| $MgCl_2$ | Teknova | M0304 | M030427E1401 | all | 1 M | Biotechnology |
| $CaCl_2$ | Amresco | E506-500mL | 0982C098 | all | 0.95-1.05 M | Biotechnology |
| HCl | Fisher Scientific | SA48-1 | 175004 | 1, 2, 3 | 0.999 N | Certified |
| HCl | Fisher Scientific | SA48-1 | 188768 | 4 | 1.003 N | Certified |


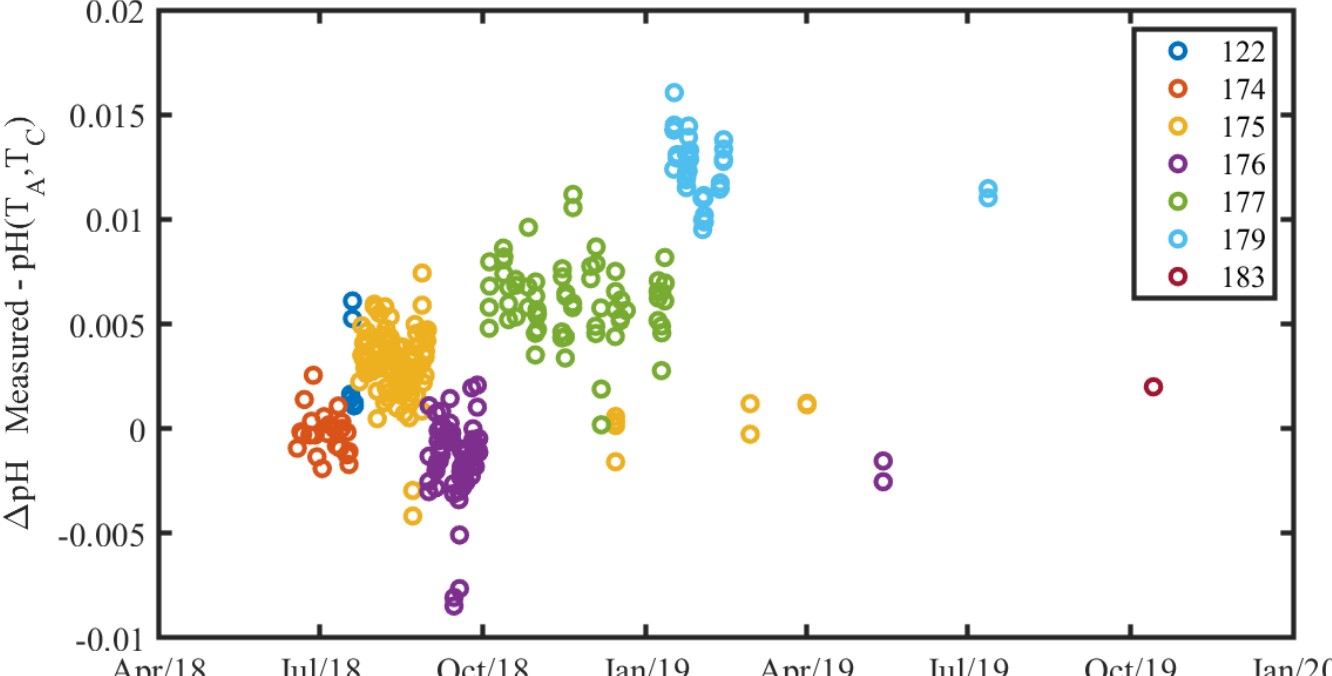


**Fig. A1: A timeseries of the residual between measured and calculated CRM pH throughout the experiment. Marker color**
**denotes CRM batch number. There is a clear variability between measured and calculated pH, which typical of CRM**
**batches (Andrew Dickson, *pers. comm.*). There was no observable systematic drift in the pH system during the experiment.**
**The mean standard deviation of pH measurements within a CRM batch is 0.0016, which is comparable to the 0.0019**
**reported in Bockmon & Dickson (2015). The same 760 nm absorbance wavelength outlier removal procedure used for tris**
**measurements was applied to CRM measurements.**

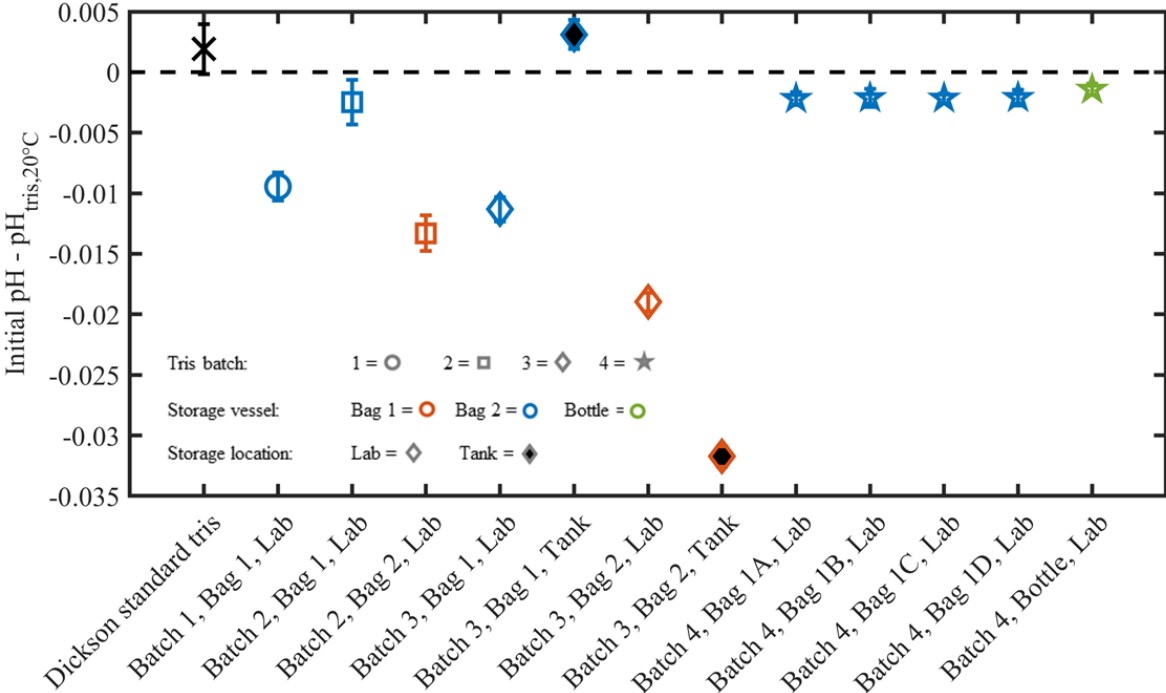


Fig. A2: The initial pH residual of each tris bag or bottle measured in this experiment. The initial pH is reported as a residual from the calculated pH at 20 °C. The initial pH was measured directly for tris batch 4 and extrapolated for tris batches 1-3. Additionally, 2 bottles of Dickson standard tris (show by the black "X") were measured on 12/10/2018. The zero black dashed line is the calculated pH of tris at 20 °C, based upon the measured reagent concentrations (DelValls and Dickson, 1998).

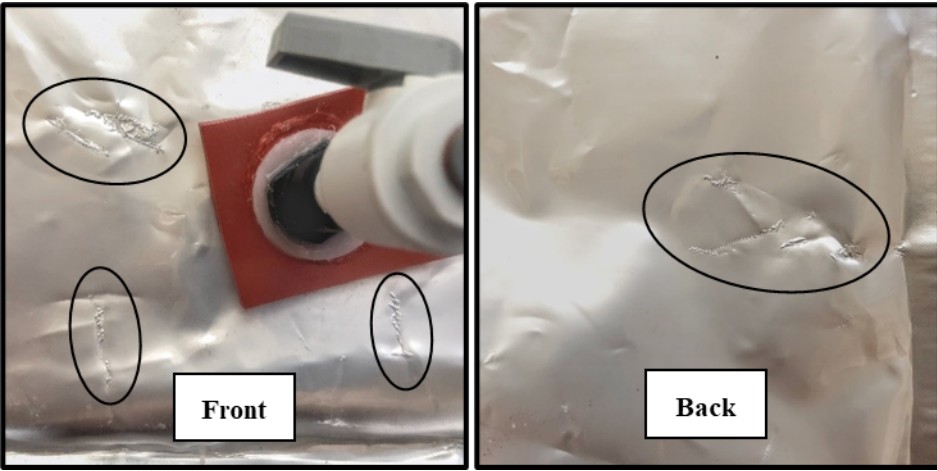

Fig. A3. The ovals indicate marks on the exterior of "Batch 2, Bag 1, Lab". These marks appear to be damage to the interior metallic layer, possibly due to creasing of the bag. These marks were not present on any other bag used in this study.

**Author contribution**

WW performed formal analysis, visualization, and writing – original draft preparation. KS and TW contributed to investigation and writing – review & editing. PB, YT, and TM contributed to funding acquisition, conceptualization, formal analysis, and writing – review & editing.

**Competing interests**

The authors declare that they have no conflict of interest.

**Data availability**

pH and $C_T$ data are available via the UC San Diego Library Digital Collections at https://doi.org/10.6075/J0QC022G (Wolfe et al., 2021).

**Acknowledgements**

We thank May-Linn Paulsen and Andrew Dickson's laboratory for sharing their tris expertise throughout this project. We thank the National Science Foundation Ocean Technology and Interdisciplinary Coordination (NSF-OTIC 1736905 and NSF-OTIC 1736864) and the David and Lucile Packard Foundation for supporting this work.

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
