# Peer review of "Technical Note: Stability of tris pH buffer in artificial seawater"

_Ocean Science, 2020_

## Author Comment (AC1)

Ocean Sci. Discuss., referee comment RC1
https://doi.org/10.5194/os-2020-120-RC1, 2021

[Figure]

**Comment on os-2020-120**

Jens Daniel Müller (Referee)

Referee comment on "Technical Note: Stability of tris pH buffer in artificial seawater stored in bags" by Wiley H. Wolfe et al., Ocean Sci. Discuss., https://doi.org/10.5194/os-2020-120-RC1, 2021

**General comments**

**Short summary:** This technical note introduces the use of gas tight bags for the storage of pH buffer solutions, as an alternative to the storage in bottles. The authors demonstrate - through comparison pH measurements with the spectrophotometric method - that the decrease of the buffer pH can be limited to <0.01 over a period of one year and under laboratory conditions. This is an important finding indicating the potential suitability of such storage bags for repeated calibration measurements of pH instruments during field deployments. The observed pH decrease is attributed to the accumulation of $CO_2$ and likely sources of $CO_2$ are discussed.

**Overall quality:** The technical note is well structured and clearly written. Appropriate references are included and the most important information to ensure reproducibility of the findings are given. This manuscript certainly presents an important contribution to the field and is well placed as a technical note in the Ocean Science journal. I would like to thank the authors for their effort to perform, evaluate and present this study.
We thank the reviewer for their helpful comments and suggestions.

Despite the overall high quality of this study, following main aspects could profit from major revisions:

(1) Spectrophotometric pH measurements performed on Certified Reference Materials reveal a spread that is larger than the repeatability that could be achieved with the instrument used (Carter et al., 2013). I wonder why this is the case and if it has an implication for the interpretation of the measurements on the buffer solutions.
The figure has been updated slightly to -use the same 760 nm as with the tris. To address these comments the figure caption has been update as follows:

"Fig. A1. A timeseries of the residual between measured and calculated CRM pH throughout the experiment. Marker color denotes CRM batch number. There is a clear variability between measured and calculated pH, which typical of CRM batches (Andrew Dickson, *pers. comm.*). There was no observable systematic drift in the pH system during the experiment. The mean standard deviation of pH measurements within a CRM batch is 0.0016, which is comparable to the 0.0019 reported in Bockmon & Dickson (2015). The same 760 nm absorbance wavelength outlier removal procedure used for tris measurements was applied to CRM measurements."

(2) It should be emphasized that the most pronounced pH changes occurred in a damaged bag. As this happened even under laboratory conditions, it appears likely that the structural integrity of the bags will represent a major challenge for successful field

deployments of the bags, in particular in the sometimes harsh conditions of the coastal zone. This aspect is mentioned in the discussion, but I think it could be emphasized a bit more, in order to avoid that the approach is considered "ready-to-go" without further testing.

To address these comments the following text has been added to the Results and Discussion as well as the Conclusions:

"The damage appears to be a break in the metallic bag layer, potentially caused by creasing or pinching of the bag when handling. This observation highlights the importance of maintaining bag integrity, particularly during use in the field. A successful two-week field deployment has been conducted using the tris bags described here and a modified SeapHOx in a shallow, coral reef flat (Bresnahan et al. 2021). This two-week deployment was significantly shorter than the year of storage described here and further field testing in longer deployments in varied environments are required before widespread use of this technology."

"While valuable at the current stage of development (as demonstrated by, e.g., Lai et al. (2018) and Bresnahan et al. (2021)), further development would ideally result in a commercially available bag and filling procedure that can yield a rate of pH change less than the climate threshold of 0.003 per year. This will require further tests to identify the source of $CO_2$, gas exchange or microbial respiration, as well as steps to reduce or eliminate these sources."

(3) Although the essential information is already covered throughout the manuscript, I would like to encourage the authors to summarize specific instructions for potential users

of this approach in a dedicated "Recommendations" section. This section could be guided along three questions: What are the core requirements (material, fittings, etc) of gas tight bags to be used for pH buffer storage? What needs to be considered for proper handling of the bags (cleaning, filling, storage, etc)? Which measures could be taken to ensure that the bags perform well under in situ conditions?

We have added recommendations to the end of Results and Discussion:

"These results suggest that when bags are carefully handled prior to and after filling, tris pH changes are small over time. Specific recommendations for further work include: bags must be handled with care and enclosed in protective containers to prevent damage, bags must be rinsed with tris prior to filling, and additional testing is merited to determine sources of and methods to reduce contamination, such as acid washing"
* * *
**Specific comments**

L. 11: Usually, the buffers are prepared as "equimolal" not "equimolar" solutions (DelValls and Dickson, 1998). Please check and - if applicable - correct throughout ms.
Corrected to equimolal throughout.

L. 13: I wouldn't consider the tris batches "experimental conditions", but rather replicates of the same experiment. Consider removing it from the list here.
Removed.

L. 15: It appears equally (or even more) important to inform the reader about the range rather than the mean of pH changes that the buffers are likely to experience. This information is indicated by the CI (± 0.0008 yr−1), but surprisingly this CI seems to disagree with the range given in the caption of Figure 3 where the upper and lower bounds are given as -0.0028 and -0.0091 pH per year. Can you clarify how the CI of the slope relates to the upper and lower bounds, and include the upper and lower bounds in the abstract?
The calculation and reporting of confidence interval were corrected to address this comment. The reported upper and lower bounds are now the sum of the confidence in the intercept and the slope at day 365. Text was added both to the figure caption and

manuscript to clarify:

"The upper and lower bounds of ΔpH at t = 365 days, -0.0042 and -0.0076, are important to consider when utilizing this bagged storage method of tris. These bounds provide the broadest expected range in pH change over a year of storage, and include both the intercept and slope confidence intervals."

L. 33: Uncertainty thresholds are listed in the wrong order. It should be 0.02 for the weather and 0.003 for the climate goal (Newton et al., 2015).
Fixed.

L. 37: The expression "roughly once per decade throughout most of the ocean" sounds to me as if only one major ocean cruise can be conducted per decade. Maybe change to "decadal reoccupations of a few major sections per ocean basin".
Fixed.

L. 81ff.: When I understood correctly, bag type 1 was custom made and type 2 is commercially available. Is this correct and can it be clarified early in chapter 2?
The text "was custom made" and "commercially available" were added to the bag type descriptions.

L. 99: When I understand it correctly, the unit error affected only the HCl concentration but not the TRIS concentration. If this is the case, then the TRIS/TRISH+ ratio was not exactly 1:1. Please clarify this, and also revise the use of the term "equimolal" accordingly.
Your understanding is correct. The following sentences were added to clarify this point:

"This unit error resulted in a tris:trisH+ of 1:0.97 that slightly differs from the 1:1 of truly equimolal tris. As this ratio is nearly equimolal, the term "equimolal" will continue to be used throughout this study.

L. 100: Please include the degree of purity of the reagents, where possible. (See also comment on l. 223 below).
The details about the specific reagents used in this study has been removed from the Methods section and added to Appendix A organized in a table format (Table A1). This table includes some additional information, such as chemical grades.

L. 107-122: The information in text and Table 1 appear redundant. Maybe the text could be restricted to general explanations of procedures, whereas the table could cover the specific routines for each test.
This section was condensed to avoid duplicate information between the text and Table 1.

L. 118: The description of the CT measurements would profit from more technical details. Did you need to adjust the method to measure the comparably low CT concentrations in the buffer solutions? Were CRM measured along with the buffer samples? If yes, please indicate the batch.
The following has been added:

"This IR measurement system is capable of measuring relatively low CT without requiring method adjustment and has been used to make near zero CT measurements (Paulsen and Dickson, unpublished data). CT measurements were made on CRMs (Batch 179 & 183)."

L. 133: The temperature dependence of the TRIS buffer applied here strictly refers to the equimolal buffer composition, whereas some deviations need to be expected for the buffer composition used in this study. Can you estimate how large this difference might be? Does the dependence of pHspec,Tc on temperature agree with the expected temperature dependence of the buffer solutions?
We add the following text to our description of the temperature correction in order to address this concern:

"This adjustment assumes that any potential difference in ∂pH/∂T between that corresponding to equimolal tris and that corresponding to our 1:0.97 tris:trisH+ ratio has a negligible effect over the small temperature range observed."

L. 136: The time series of CRM measurements does not show a systematic drift, but a spread (~ 0.03 pH units) that is about an order of magnitude higher than the precision that can be achieved with the instrument used, and also larger than the good agreement between measured and calculated TRIS pH in this study would suggest (compare l. 185 - 192). Measured pH values appear clustered by CRM batch and/or measurement time. Would you have an explanation for this and an idea how this would impact your findings?

The observed variability between measured pH vs the pH calculated from CT and alkalinity is in the range of expected values (Andrew Dickson, pers. comm.). This should not affect our findings, as the important metric is the repeatability within a single CRM batch, as this demonstrates the repeatability of our pH spec measurements. The average std dev of pH of a single CRM batch we obtained (0.0016) is similar to that obtained in A. Dickson's lab (0.0019 (Bockman & Dickson, 2015)). Since no trend in CRM pH was observed for each batch of CRM, this indicates that our spec pH measurements were stable and repeatable throughout the experiment. The caption of Fig. A1. has been updated to address this comment:

"Fig. A1. A timeseries of the residual between measured and calculated CRM pH throughout the experiment. Marker color denotes CRM batch number. There is a clear variability between measured and calculated pH, which typical of CRM batches (pers. comm. Andrew Dickson). There was no observable systematic drift in the pH system during the experiment. The mean standard deviation of pH measurements within a CRM batch is 0.0016, which is comparable to the 0.0019 reported in Bockmon & Dickson (2015). The same 760 nm absorbance wavelength outlier removal procedure used for tris measurements was applied to CRM measurements."

L. 137: I'm a bit sceptical about the approach to correct for dye impurities. First, I'm wondering if this correction is required at all, as the aim is to track pH changes (or better pH stability) over time and therefore the pH-dependent impact of dye impurities should be almost identical for all buffer measurements. More importantly, the comparison measurement of pure vs impure dye made on seawater solutions should also be affected by the pH perturbation of the dye addition. This pH perturbation is related to the pH of the stock solution and can be different for the two stock solutions used here. Did you minimize this pH perturbation by adjusting the stock solution pH to the sample pH, or correct for it by extrapolating your measurements to zero dye concentration? If not, I'm afraid your correction term in Eq. (2) might be impacted. Please revise this approach.

To address the reviewer's comments, the following text was added:

"Varying ratios of tris:trisH$^+$ were used to obtain different solution pH, and to buffer any changes in pH during the experiment, which negates the need for dye perturbation corrections in this characterization."

"All subsequent pH$_{spec}$ measurements in this study were conducted with impure dye and are reported with this dye impurity correction (Eq. 2) applied. The correction adjusted the reported pH by 0.0093 ± 0.0002 (mean ± standard deviation, n = 126). No dye perturbation correction was used (a correction for a change in pH caused by the addition of the dye). As the high buffering capacity of tris, in combination with a dye adjusted to a pH similar to that of tris, results in a negligible change in measured pH."

We chose to include the dye impurity correction to assess the accuracy of the chemical preparation (as in Fig. A2). The correction affects the difference between the measured initial pH and the calculated pH of tris, while not significantly impacting the slope.

L. 145: What do you mean with "normal practices"? Is this a standard operating procedure, or a threshold defined in your lab? Can you provide a reference?
The phase "Following normal practices" was removed to avoid confusion.

L. 198: The estimates of the upper and lower bounds should be given more weight. From a users perspective and for the application of this storage solution without regular pH test measurements, the likely range of pH changes seems even more important the average rate of change! Please explain in the main text, how these bounds must be interpreted in contrast to the 95% CI of the slope.
The following sentence has been added to clarify this point:

"The upper and lower bounds of ΔpH at t = 365 days, -0.0042 and -0.0076, are important to consider when utilizing this bagged storage method of tris. These bounds provide the broadest expected range in pH change over a year of storage, and include both the intercept and slope confidence intervals."

L. 204: The consequence of bag damage deserves more attention, in particular with respect to the use of bags under in situ conditions. How can this damage happen even under laboratory conditions, and more importantly, how can it be avoided?
This section has been expanded to discuss the possible cause of the damage in lab conditions as well as a now published field deployment of a tris bag. The following sentences have been added to the section.

"The damage appears to be a break in the metallic bag layer, potentially caused by creasing or pinching of the bag when handling. This observation highlights the importance of maintaining bag integrity, particularly during use in the field. A successful two-week field deployment has been conducted using the tris bags described here and a modified SeapHOx in a shallow, coral reef flat (Bresnahan et al. 2021). This two-week deployment was significantly shorter than the year of storage described here and further field testing in longer deployments in varied environments are required before widespread use of this technology."

L. 214: Could your interpretation "that the drift in tris pH was primarily driven by an increase in CO2" also be supported by the change of CT over time? Did you see a consistent increase of CT? I assume this should be the case, due to the fact that pH is decreasing over time and CT and pH seem to be correlated. However, an explicit statement about this would not hurt.
"and CT" explicitly added to the sentence.

L. 216: Can you please describe in more detail what you mean with an "ad hoc acid–base equilibrium model of seawater including tris in addition to the CO2 and other minor acid–base systems"? I've an idea what you mean but it is not entirely clear to me.
The acid-base model description has been expanded to provide more information:

"The theoretical change in tris-artificial seawater (ASW) pH due to an increase in CT is straightforward to calculate, since both tris and CO2 acid-base equilibria are well-characterized in seawater and ASW media. The pH is calculated for tris-ASW + CT using an equilibrium model following the approach described in Chapter 2 of Dickson et al. (2007) for the case of known alkalinity and CT. In the case of ASW, the seawater equilibrium constants for CO2 are appropriate because minor ions present in seawater and not ASW do not appreciably affect the CO2 equilibrium constants (particularly when the goal is to compute relative changes in pH) as the ionic background of ASW is closely matched to that of seawater at salinity = 35. In our model, minor acid-base species important to seawater alkalinity but not present in ASW (borate, phosphate, silicate, fluoride) are set to zero. The definition of total alkalinity is modified to include the tris acid-base system following the definition of acid-base donor/acceptor criteria given by Dickson (1981): tris is assigned as a level-1 proton acceptor and tris-H+ is at the zero level. Thus, in our model, tristot = 0.08 molal and alkalinity = 0.04 molal and CT is a variable. An algorithm (see Annexe 1 in Dickson et al. (2007)) is then used to find the root of the alkalinity equation in its residual form by solving for pH."

L. 223: Respiration of organic matter is proposed as one potential source for the accumulation of CT. Could you try to relate the amount of accumulated CT to the size of potential sources? Would it be possible to give a conservative estimate of how much

organic material could cover the inner wall of the bags? Which quantity of organic matter must be expected to be contained in the reagents used to produce the buffer solutions? Is tris itself - which is also routinely used in biological experiments to stabilize pH - likely to be respired? I think a bit more detailed discussion to this end would help to identify how the accumulation of CT can be prevented in the future.

We are unable to estimate organic matter contamination, but we attempt to address the reviewer's questions in the subsequent paragraph as well as with the following added text:

"Beyond removing organics on the bag surfaces, care should be taken to avoid introducing organic contaminates into the tris during the solution preparation and bag filling procedures to minimize future respiration."

L. 280: I was not able to access the data at UC San Diego Library Digital Collections through the doi, nor through a keyword search. Please make sure that the data are correctly uploaded and accessible.

Data are now public through the UCSD Library. The clarification "at https://doi.org/10.6075/J0QC022G" has been added to the data availability section
* * *
**Technical corrections**

L. 13: I think the wording "flexible bag" is a pleonasm. The word "flexible" can be removed here and throughout the ms.

Three uses of "flexible" were removed throughout the manuscript.

L. 17: The explicit drift rate can be removed here in order to avoid repetition of the same number within the abstract.

Removed.

L. 18 Consider replacing "value" by "potential", as in situ applicability has not yet been demonstrated.

Replaced.

L. 33: A second edition of this document was made available by Newton et al. (2015). Please update the reference.

Updated.

L. 56: Rephrase "deep, comparatively stable ocean" to "deep ocean with comparatively stable pH" or similar

Fixed.

L. 62: Replace ", one or more times" by "repeatedly"

Replaced.

L. 65: Include reference Papadimitriou et al., (2016)

Added.

L. 78: Replace "for CO2" with "for oceanic CO2 measurements"

Replaced.

L. 91: Introduce abbreviation HDPE

Changed to "high density polyethylene" as HDPE abbreviation is not used again.

L. 153: Data availability statement can be removed here, as it is given in a separate section below.

Removed.

L. 156: To my impression, the term "drift" is more frequently used to describe the change of a measured value due to changes in instrument performance, i.e. instrument drift.

Here, you are referring to real pH changes of the solution. Please consider rephrasing to "A near-linear decrease of pH was …" or similar.

"drift" describing change in tris pH was replaced with "decrease" or "change".

L. 158: Replace "is" by "was"

Fixed.

L. 158: Does Table 2 report measured values at t = 0 when those are available, or always the intercept of the fitted regression model? Text and table caption read contradictory in this respect.

The clarification "The reported intercept is the regression intercept, when initial pH measurements are available, they differ by less than 0.0003 from regression intercept." Has been added to Table 2 description.

Fig. 2: Showing one type of symbol and the corresponding legend per panel appears redundant. I recommend to use either the same symbols and color in all panels and keep only the descriptive label in each panel, or replace the individual legends by three joined legends indicating what the symbols, color and fill represent. Overall, axis labels and text appear small in this figure. Please try to increase text size and - if necessary - make use of the full page height to plot the panels. Consider starting the caption with "Individual time series of measured pH in tris buffer solutions ..."

The individual legends were replaced with just the descriptive labels. Additionally, a marker description has been added to the bottom right of the figure. The figure dimensions have been increased. The caption was updated following your suggestion.

Fig. 3: Consider starting the caption with "Combined time series of measured pH in tris buffer solutions ..."

Caption updated.

L. 201-202: The sentences "By definition … small magnitude" could be removed.

Removed.

L. 220: Consider replacing "has been designed to" with "is known to"

Replaced.

L. 250: Please revise placement of "the bag" in "studies successfully used bag type 2 submerged the bag in seawater for less time"

Reworded.

L. 260: For consistency, remove "purportedly" here, or also include it in the abstract.

"Purportedly" added to the abstract.

Supplementary materials: According to the manuscript preparation guidelines, the supplementary figures of this study should be placed in appendices. (Copied from the Ocean Science website: "Additional figures, tables, as well as technical and theoretical developments which are not critical to support the conclusion of the paper, but which provide extra detail and/or support useful for experts in the field and whose inclusion in the main text would disrupt the flow of descriptions or demonstrations may be presented as appendices." and "Supplementary material is reserved for items that cannot reasonably be included in the main text or as appendices. These may include short videos, very large images, maps, CIF files, as well as short computer codes such as matlab or python script.")

All information previously in supplementary materials has been moved into Appendix A.
* * *
**References used in this review**

Carter, B. R., Radich, J. A., Doyle, H. L., and Dickson, A. G.: An automated system for spectrophotometric seawater pH measurements: Automated spectrophotometric pH measurement, Limnol. Oceanogr. Methods, 11, 16–27, https://doi.org/10.4319/lom.2013.11.16, 2013.

DelValls, T. A. and Dickson, A. G.: The pH of buffers based on
2-amino-2-hydroxymethyl-1,3-propanediol ('tris') in synthetic sea water, Deep Sea Res.
Part Oceanogr. Res. Pap., 45, 1541–1554,
https://doi.org/10.1016/S0967-0637(98)00019-3, 1998.

Newton, J., Feely, R., Jewett, E., Williamson, P., and Mathis, J.: Global Ocean Acidification
Observing Network: Requirements and Governance Plan. Second Edition, 2015.

Papadimitriou, S., Loucaides, S., Rérolle, V., Achterberg, E. P., Dickson, A. G., Mowlem,
M., and Kennedy, H.: The measurement of pH in saline and hypersaline media at sub-zero
temperatures: Characterization of Tris buffers, Mar. Chem., 184, 11–20,
https://doi.org/10.1016/j.marchem.2016.06.002, 2016.

Dickson, A. G., Sabine, C. L., and Christian, J. R.: Guide to Best Practices for Ocean CO2
Measurements, PICES Special Publication 3, North Pacific Marine Science Organization,
Sidney, British Columbia, 191 pp., 2007.

Bockmon, E. E., and Dickson, A. G.: An inter-laboratory comparison assessing the quality
of seawater carbon dioxide measurements, Mar. Chem., 171, 36-43,
https://doi.org/10.1016/j.marchem.2015.02.002, 2015.

---

## Author Comment (AC2)

Ocean Sci. Discuss., referee comment RC2
https://doi.org/10.5194/os-2020-120-RC2, 2021

[Figure]

**Comment on os-2020-120**

Anonymous Referee #2

Referee comment on "Technical Note: Stability of tris pH buffer in artificial seawater stored in bags" by Wiley H. Wolfe et al., Ocean Sci. Discuss., https://doi.org/10.5194/os-2020-120-RC2, 2021

Oceanic pH is affected globally by climate change and on smaller scales by a range of physical, chemical and biological processes. Quantifying and understanding these changes is dependent on an effective quality assurance system for oceanic pH measurements. This manuscript contributes to this development by assessing the stability of bagged pH buffers deployed in seawater: use of these bagged buffers provides the ability to include reference standards in pH measurement campaigns in situ. I recommend that the authors address the following points before publication:
We thank the reviewer for their helpful comments and suggestions.

Line 33: the "climate" and "weather" uncertainty levels are the wrong way round.
Fixed.

Lines 104-120: set the definitions of Tests 1, 2 and 3 in separate subparagraphs
The information in these paragraphs was cut down as per the suggestion by another reviewer to avoid duplication between the text in Table 1.

Line 139: state the sources of the impure and pure dyes
The source or purified dy as been added to Line 138:

"impure dye ($pH_{impure}$; from Aldrich, lot MKBH6858V) and purified dye ($pH_{pure}$; from Robert Byrne's Lab, University of South Florida)"

Line 216: reference to an "ad hoc speciation model" is unacceptably vague: if a speciation model is to be used then full details should be given. In this case I advise strongly against using a model since even at the standard physical chemistry temperature of 25°C we lack an adequate model of Tris chemistry in seawater. The correlation shown in Figure 3 is good evidence that $CO_2$ is the culprit: modelling Tris buffer chemistry does not provide additional evidence given the uncertainties in the available models.
The acid-base model description has been expanded to provide more information.

"To assess if the change in pH was driven by the addition of CO2, the final pH and available CT measurements were compared with a model described here. The theoretical change in tris-artificial seawater (ASW) pH due to an increase in CT is straightforward to calculate, since both tris and CO2 acid-base equilibria are well-characterized in seawater and ASW media. The pH is calculated for tris-ASW + CT using an equilibrium model following the approach described in Chapter 2 of Dickson et al. (2007) for the case of known alkalinity and CT. In the case of ASW, the seawater equilibrium constants for CO2

are appropriate because minor ions present in seawater and not ASW do not appreciably affect the CO2 equilibrium constants (particularly when the goal is to compute relative changes in pH) as the ionic background of ASW is closely matched to that of seawater at salinity = 35. In our model, minor acid-base species important to seawater alkalinity but not present in ASW (borate, phosphate, silicate, fluoride) are set to zero. The definition of total alkalinity is modified to include the tris acid-base system following the definition of acid-base donor/acceptor criteria given by Dickson (1981): tris is assigned as a level-1 proton acceptor and tris-H+ is at the zero level. Thus, in our model, tristot = 0.08 molal and alkalinity = 0.04 molal and CT is a variable. An algorithm (see Annexe 1 in Dickson et al. (2007)) is then used to find the root of the alkalinity equation in its residual form by solving for pH."

The authors conclude that bag storage has been shown to be adequate, and do not propose any further development. I consider this conclusion to be premature for two reasons. First, the commercial bag that was tested delaminated when stored in seawater, so that the only bag shown to perform adequately in seawater was Bag 1, which appears to have been made in the authors' laboratory. If the use of bagged buffers is to become routine for in situ pH measurements, then bags that meet the driftspecifications need to be commercially available: I consider that this point should be made in the conclusions. Second, the authors conclude that two factors may contributeto the observed decline in buffer pH: leakage of $CO_2$ into the bag; and production of $CO_2$ by respiration. In order to optimise bag design and the cleaning and filling procedure, experiments should be undertaken to identify the major cause of $CO_2$ This should also be stated in the conclusions.

To address the reviewer's comment, the following sentences have been added to the conclusions.

"While valuable at the current stage of development (as demonstrated by, e.g., Lai et al. (2018) and Bresnahan et al. (2021)), further development would ideally result in a commercially available bag and filling procedure that can yield a rate of pH change less than the climate threshold of 0.003 per year. This will require further tests to identify the source of CO2, gas exchange or microbial respiration, as well as steps to reduce or eliminate these sources."

---

## Author Response (AR2)

**Report #1**

Submitted on 11 May 2021
Referee #1: Jens Daniel Müller, jensdaniel.mueller@usys.ethz.ch

**Anonymous during peer-review:** Yes **No**

**Anonymous in acknowledgements of published article:** Yes **No**

**Recommendation to the editor**

| | |
|---|---|
| **1) Scientific significance**
 Does the manuscript represent a substantial contribution to scientific progress within the scope of this journal (substantial new concepts, ideas, methods, or data)? | Excellent **Good** Fair Poor |
| **2) Scientific quality**
 Are the scientific approach and applied methods valid? Are the results discussed in an appropriate and balanced way (consideration of related work, including appropriate references)? | Excellent **Good** Fair Poor |
| **3) Presentation quality**
 Are the scientific results and conclusions presented in a clear, concise, and well structured way (number and quality of figures/tables, appropriate use of English language)? | Excellent **Good** Fair Poor |

For final publication, the manuscript should be

**accepted as is**

**accepted subject to technical corrections**

accepted subject to **minor revisions**

reconsidered after **major revisions**

 I am willing to review the revised paper.

 I am **not** willing to review the revised paper.

**rejected**

**Suggestions for revision or reasons for rejection (will be published if the paper is accepted for final publication)**

The authors did a good job in implementing the comments made by both reviewers, with one substantial exception being the revision of the confidence interval of the observed pH changes over time.

We thank Dr. Müller for the careful re-review of the manuscript and we address the comments and suggestions below.

In the first submitted version of the manuscript, the authors stated that:
"A linear regression on all pH measurements [...] has a slope of –0.0058 ± 0.0008 yr–1 (mean ± 95% C.I.)."

In the revised version of the manuscript, the CI of the slope was changed and additional information is given:
"A linear regression on all pH measurements [...] has a slope of –0.0058 ± 0.0011 yr–1 (mean ± 95% C.I.). The upper and lower bounds of ΔpH at t = 365 days, -0.0042 and -0.0076,are important to consider when utilizing this bagged storage method of tris. These bounds provide the broadest expected range in pH change over a year of storage, and include both the intercept and slope confidence intervals."

There are several follow-up questions and recommendations to this change, which I suggest the author take into account:
- Why did the numeric value of the CI increase, while the grey shaded area in Fig. 3 is now narrower than in the first version?
- Which version (first submission or resubmission) of the numeric CI value in the text and the CI interval in Fig. 3 are correct?
- The fact that the grey shaded area in Fig. 3 is now narrower appears contradictory to the statement that it represents the "broadest expected range in pH change".
The confidence interval in the first submission was calculated from the mean of replicate measurements. The confidence interval in the resubmission was calculated from all individual measurements, resulting in a slightly larger confidence interval.

The resubmission has the correct value of the numeric CI in the text and CI in Fig. 3.

In the original submission, we misused the confidence interval function in MATLAB resulting in an incorrect exaggeration of the illustrated wedge and a mismatch between the reported numeric range and the illustrated range in the wedge in Fig. 3. We corrected this mistake which results in the new values as reported and illustrated in Fig. 3 in the resubmission. This correction now provides the broadest expected range in pH.

- Please clarify how the CI of the slope and the intercept were combined to derive the upper and lower bounds.
Both the slopeCI and interceptCI contribute to the upper and lower bounds.

The following text is added to clarify:
These bounds provide the broadest expected range in pH change over a year of storage and include both the intercept and slope confidence intervals ($slope_{CI}$ and $intercept_{CI}$, respectively). For example, the upper bound of ΔpH at t = 365 days is calculated as:
$$upper\ bound = (slope + slope_{CI}) * 365 + intercept + intercept_{CI}.$$

- Finally, I repeat my suggestion to include the information contained in the upper and lower

bound estimates also in the abstract. It is likely among the most relevant piece of information for many readers.

Added the following text to the abstract:

The upper and lower bounds of expected pH change at t = 365 days, calculated using the averages and confidence intervals of slope and intercept of measured pH change vs. time data, were -0.0042 and -0.0076 from initial pH.
* * *
Additional technical comments are:

L.136ff: "To account for pH-dependent errors from impurities in unpurified mCP, a pH-dependent correction factor was determined based on the protocol outlined in Takeshita et al. (in reiew). Briefly, pH of seawater was measured subsequently using impure dye (pHimpure; from Aldrich, lot MKBH6858V) and purified dye (pHpure; from Robert Byrne's Lab, University of South Florida (Liu et al., 2011)) over a range of pH between 7.4 to 8.2 at approximately 0.2 intervals. Varying ratios of tris:trisH+were used to obtain different solution pH ..."

With the new information included in the revision it is now clear that tris buffered solutions were used to determine the impact of dye impurities, and not seawater solutions. Thus, I agree with the applied correction procedure. However, please replace:
- "pH of seawater was measured" with "pH of tris buffered artificial seawater solutions was measured".

Replaced with "Briefly, pH of natural seawater with different ratios of added tris:trisH$^+$ was measured…"
- "reiew" with "review".

Replaced.
- "Lab" with "laboratory"

Replaced.

In Figures 2 and A2, I suggest to avoid redundant information to make the figures more easy to read.

Fig 2:
- Remove labels in plot, as they are redundant with the color/shape.

We disagree with the reviewer on this point, believing that they make the plot easier and quicker to understand. We have changed the label and slope order to make it easier to read.
- Remove "- shown by marker XXX" legend

Removed.
- In the caption, consider changing "Bag type 1 is shown in blue (light blue for the 185damaged bag of type 1), 2 in orange and bottle in green. Tris batch 1 is depicted as circles, 2 as squares, 3 as diamonds and 1864 as stars. Storage location in tank has a black fill and lab symbols have no fill." to "Storage vessel is indicated by color, storage location by fill, and Tris batch by color"

Replaced with "Tris batch is indicated by shape, storage vessel by color, and storage location by fill."

Fig A2:
- Shapes and fill in the legend are not readable. Consider using the same legend as in Fig. 2.
Replaced with Fig. 2 legend.
* * *
Thanks again to all authors for developing and testing this new approach to perform quality control of pH measurements under field conditions. I hope your findings will be adapted rapidly.
Thank you for all the feedback on the manuscript.